# Recommended Separation Distances for 1.3 Ammunition and Explosives

**Clint Guymon** [1,2,*] **, Ming Liu** [3] **and Josephine Covino** [4]

1 Safety Management Services, Inc., West Jordan, UT 84088, USA
2 Department of Chemical Engineering, Brigham Young University, Provo, UT 84602, USA
3 Naval Facilities Engineering and Expeditionary Warfare Center, Port Hueneme, CA 93043, USA
4 Department of Defense Explosives Safety Board, Alexandria, VA 22350, USA
* Correspondence: clint.g@byu.edu

**Abstract:** Separation Distances are used throughout the world to protect people and assets from the potential hazardous effects from propellants, explosives, and pyrotechnics. The current separation distances for Hazard Division (HD) 1.3 substances and articles used in the United States, in some cases, may not adequately protect against the effects from heat flux and debris when those substances and articles are ignited in a confined structure. Multiple tests in such a confined scenario with HD 1.3 substances have shown that the heat flux and debris hazards could result in injury at distances beyond the current specified explosives safety separation distance (ESSD). Herein are the recommended ESSDs for confined as well as unconfined HD 1.3 articles and substances based on the analysis of hundreds of tests. Recommended ESSDs include a smaller value for unconfined quantities less than 145 kg and ESSDs that are consistent with NATO distances for confined substances and articles.

**Keywords:** propellant; HD 1.3; safe separation; quantity distance

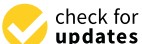

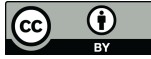

## 1. Introduction

The safe manufacturing and staging of energetic substances and articles requires stand-off distances to reduce the severity of the effects in an ignition scenario. The United States Department of Defense has specified those stand-off distances in the Defense Explosives Safety Regulation 6055.09 (DESR) [1], which is used as the standard throughout the United States. Within that document are stand-off distance guidelines for the different divisions of energetics including Hazard Division (HD) 1.1, 1.2, and 1.3. The stand-off distance is known as the Explosives Safety Separation Distance or ESSD. The ESSD to an inhabited building (i.e., Inhabited Building Distance, IBD) is the distance required to protect structures occupied by people both inside and outside Department of Defense property lines. The protection is for blast pressures and effects, debris or fragments, and thermal hazards. This paper is focused on hazards associated with HD 1.3 substances and articles.

Multiple scenarios including accidents and experimental test results (see, for example, References [2–4]), have shown that the current HD 1.3 ESSD in the DESR may not adequately protect against hazards from burning propellants, explosives, or pyrotechnics when in a structure that does not provide adequate venting (i.e., a confined scenario). Boggs, et al. (Ref. [2]) details 80+ accidents; one of which is the Milan 2004 storage magazine event (primarily with HD 1.3 substances) that resulted in a hazardous fragment density beyond the IBD with a large portion of the magazine head wall thrown 1300 feet. The experimental testing of confined scenarios of HD 1.3 M1 propellant inside a small concrete structure results in hazardous fragment densities beyond the IBD for multiple test scenarios as reported with detailed mappings of debris fields and heat flux measurements in Reference [3]. Indeed, the DESR 6055.09 acknowledges significant hazards from the confined burning of HD 1.3 substances and articles, as mentioned in Section V1.E8.4.1.3 of DESR 6055.09:

Where there is minimal venting and structural containment (extreme confinement), an [explosion] of the HD 1.3 may occur with effects similar to those of an HD 1.1 explosion. For example, HD 1.3 AE is considered HD 1.1 (mass explosion) for QD purposes when stored in underground chambers.

Despite the above quoted language, the specification of "minimal venting" or "structural containment similar to an underground chamber" is insufficient to protect against significant injury from accidental ignition of HD 1.3 substances and articles. Or, in other words, the HD 1.3 distances in the widely used DESR do not distinguish between confined or unconfined scenarios, and assignment of the much larger HD 1.1 IBD distances, as suggested in Section V1.E8.4.1.3, may be over conservative. Over-conservative distances can significantly limit the efficient production and storage of explosives, pyrotechnics, and propellants. Herein are recommended explosives safety separation distances for unconfined and confined HD 1.3 ammunition and explosives (AE). The below analysis treats unconfined and confined scenarios of HD 1.3 or primarily burning reactions of energetic substances.

Unconfined burning allows the generated gases to freely exit the immediate area around the substance or article such that the pressure surrounding the substances or articles is not elevated significantly. Confined burning is when the gases generated are not able to freely escape due to insufficient venting. Insufficient venting results in an elevated pressure inside the structure. With elevated pressure, the burn rate increases substantially, as the burn rate is a strong function of pressure. With increasing pressure, the hot gases escape through the spaces and vents in the structure with greater speed, choking the gaseous flow and leading to a likely violent structural failure. Prior to structure failure, effects of those hot gases exiting through the vent extend well beyond a distance for an unconfined of unpressurized burn. With violent structural failure, the hazardous debris generated can approach that from a HD 1.1 event.

Determining whether a HD 1.3 burning scenario is unconfined or confined with choked or nearly choked flow requires a knowledge of the gas generation rate of the burning HD 1.3 material as well as the size and behavior of the gaseous escape routes (vents). Fast running models such as the Integrated Violence Model can be used to obtain an estimate of the internal pressure during the burning event and subsequent violence (Reference [5]). An analysis of hundreds of scenarios of unconfined and confined HD 1.3 experiments and accidents are presented here with recommended ESSDs as a function of the net explosive weight (NEW). Additionally, modeling results with the Integrated Violence Model are given. Validated modeling has augmented the experimental and accident data where the scale is very costly in obtaining experimental data.

## 2. Materials and Methods

An explosives safety separation distance, ESSD, from a substance, article, or structure with reacting material, specifically burning material, is one where an individual would not receive second degree burns and would not be exposed to hazardous debris (<79 Joules) at a density greater than one fragment per six hundred square feet. These standards are from the Defense Explosive Safety Regulation (DESR) 6055.09, as outlined in Sections V1.E8.2.2.4 and V1.E8.2.2.5 [1].

Determining the safe distance from a HD 1.3 event requires quantifying the heat flux and debris hazards. The debris generated from an event can be quantified and the distance at which the fragment count per 600 square feet decreases below 1 can be found, which is typically a calculated value. The heat flux allowed decreases with increasing exposure time, as per the equation in Section V1.E9.3.1.2 of the DESR 6055, where the allowable exposure time is per Equation (1):

$$t = 200 \cdot q^{-1.46} \tag{1}$$

In all the analysis reported on here, the maximum exposure time was assumed to be 20 s, which corresponds to a heat flux of 4.84 kW/m$^2$ or 0.116 cal/(cm$^2 \cdot$ s). If the burn time was longer than 20 s, that 4.84 (or 0.116) value was the allowable heat flux to

prevent a second degree burn. The 20 s exposure time maximum is a conservative estimate of the duration of time an individual would need to move away from the event. The distance at the allowable heat flux is the explosives' safety separation distance (ESSD) for thermal hazards.

The radiative heat flux received from a high temperature substance decreases with increasing distance. The rate of decrease can depend on the receptor's view factor of the emitting source and the atmospheric conditions between the receptor and the source. That rate of decrease is approximately proportional to the inverse square of the distance and is commonly referred to as the point source model:

$$q = \alpha / d^2 \tag{2}$$

where $\alpha$ is a constant and $d$ is the distance from the heat source. The point source model is used by first determining the $\alpha$ value from experimental or model data based on the reported values of the heat flux at the given distance $d$ or distances. With the calculated or regressed $\alpha$ value, the distance at which the allowable heat flux from Equation (1) could be expected is found. Enumerated, the methodology used here to obtain the ESSD with each set of test data of burning bulk explosive material is as follows:

1. Obtain the max allowable heat flux from the burn time of the test and Equation (1);
2. Estimate the $\alpha$ factor in the point source model given the average heat flux data (assumed constant over the duration) at various distances by taking the 75th inclusive percentile of the $q \cdot d^2$ values (the 75th percentile is used to better approximate the area of the curve where the allowable heat flux is expected);
3. Using the point source model with the estimated $\alpha$ factor and a view factor of 1, determine the ESSD that corresponds to the max allowable heat flux.

For example, 47.6 pounds of propellant was burned in the open with an average heat flux of 0.733, 0.467, 0.464, 0.212, and 0.166 cal/cm$^2$/s recorded at distances of 3.2, 4.0, 5.0, 6.4, and 8.0 m, respectively (Reference [6]). The burn time in this scenario was 15 s. Per Equation (1), the allowable heat flux is 0.141 cal/cm$^2$/s. The $\alpha$ value is 10.6E4 cal/s and the safe distance is estimated to be 8.7 m.

Table 1 summarizes the collection of references with test or modeling data compiled for over 100 tests and over 600 data points where the ESSD was calculated. The majority of the test data are from open burning of barrels of propellant as described in References [6–8]. Fireworks were also tested in various configurations as reported by TNO in Reference [9]. The compilation of data together with the references and calculations are available in a public github repository (github.com/clint-bg/publicationdata/safeSeparation).

Each test scenario consists of a collection of propellants with a given mass. That mass is then ignited and the resulting heat flux is measured at one or more distances away from the event to quantify the amount of thermal radiation. The heat flux values together with burn times are used to calculate the safe distance. Details of an unconfined test are given below as an example of the testing scenario and test analysis that were completed for each of the hundreds of tests.

**Table 1.** HD 1.3 Test and Modeling Measurements with References.

| Reference | Propellant Types | # of Tests | # of Measurements | Event Type |
|---|---|---|---|---|
| Hay J. E. et al. [6] | IMR5010, M1-8-SP, M1-8-MP, WC844, WC846, WCBlank | 49 | 206 | Unconfined: Heat Flux |
| Pape R. et al. [7] | M1, WC844 | 46 | 79 | Unconfined: Heat Flux |
| Wyssen [10] | GP11, 35mm prowder | 17 | 176 | Unconfined: Heat Flux |
| Paquet F. et al. [11] | SB1, SB2, DB1, DB2 | 16 | 48 | Unconfined: Heat Flux |
| Harmanny A. [9] | Fireworks | 16 | 16 | Unconfined: Heat Flux |
| Wyssen [12] | 20 mm powder | 6 | 120 | Unconfined: Heat Flux |
| Guymon C. G. [8] | WC814 | 1 | 4 | Unconfined: Heat Flux |
| Guymon C. G. [13] | Model | - | 30 | Unconfined: Heat Flux |

**Table 1.** *Cont.*

| Reference | Propellant Types | # of Tests | # of Measurements | Event Type |
|---|---|---|---|---|
| Williams M. R. et al. [14] | MTV | 12 | 36 | Confined: Heat Flux |
| Trinkler [15] | Cordite | 8 | 35 | Confined: Heat Flux |
| Blankenhagel P. et al. [16] | di-tert-butyl peroxide | 3 | 9 | Confined: Heat Flux |
| Farmer et al. [3] | M1 | 3 | 3 | Confined: Heat Flux |
| Joachim C E [17] | M1 | 3 | 3 | Confined: Heat Flux |
| Allain L. [18] | LB 7 T 72 | 2 | 12 | Confined: Heat Flux |
| Titan Crane Failure [19] | Rocket Propellant | 1 | 1 | Confined: Heat Flux |
| Wilson et al. [20] | Rocket Propellant | 1 | 1 | Confined: Heat Flux |
| Farmer et al. [3] | M1 | 2 | 2 | Confined: Debris |
| Farmer et al. [21] | M1 | 1 | 2 | Confined: Debris |
| Wilson et al. [20] | Rocket Propellant | 1 | 1 | Confined: Debris |
| Guymon C. G. [5] | Model | - | 25 | Confined: Debris |

## 2.1. Example of Unconfined Test Details

The previously unpublished testing of a top ignition of a single barrel of WC814 smokeless powder [8] provides a good example of the testing typical of that performed to determine the heat flux from unconfined scenarios.

A single metal barrel (diameter of 36 cm) with 27 kg of WC814 propellant with an open top was ignited at the top of the collection of propellant. Medtherm Schmidt-Boelter Series 64 heat flux gauges were used to measure the heat flux from the burning propellant at distances of 8, 15, 20, and 24 feet from the edge of the barrel oriented in a linear array with the gauges slightly offset so they are not shielded by a closer gauge. The height of the gauges coincided with the height of the barrel. The face of the gauges were oriented parallel to the barrel wall. The testing was completed at the Tooele Army Depot. The average heat flux of the 120 s burn was 3.51, 1.38, 0.84, and 0.63 kW/m$^2$ for the above distances, respectively. Those average heat fluxes were then used according to the above method to estimate the safe separation distance to prevent a second degree burn. That distance is estimated to be approximately 8 feet or 2.5 m. This is a satisfactory result, as the allowable heat flux to prevent a second degree burn for that burn duration is 4.84 kW/m$^2$, which, experimentally at 8 feet, was less than that value.

## 2.2. Modeling Details

The modeling of heat flux and debris scenarios with confined and unconfined propellants is beneficial, as the cost of completing the modeling study can be significantly less than the experimental scenarios, especially with larger masses of substances. There are many different model types but discussed and used here are empirically and theoretically based fast-running models. Fast-running models typically do not resolve the three-dimensional or positional dependent result but estimate the average result as a function of time or distance.

### 2.2.1. Heat Flux Modeling Details

COMSOL Multiphysics [13] is a commercial simulation package that was used to estimate the radiative heat flux from the burning of propellant inside barrels of different diameters. Burn rates modeled ranged from 0.5 to 70 kg/s with various barrel diameters from 36 to 160 cm. The gas flow rate from the top of the barrel was estimated from the burn rate and a gas generation rate for smokeless powder of 30 moles per kilogram. The modeled exit gas temperature from the barrel was 2800 K. Other simulation parameters such as the emissivity of the gas were adjusted once to match experimental results for the single- and multi-barrel data in References [6,8]. Once those parameters were adjusted, all that was changed was the burn rate and barrel diameters for other conditions that were not experimentally tested. The separation distances to prevent 2nd degree burns were also found according to the above described method. Figure 1 shows an example image of one of the model simulations with the temperatures shown. The barrel is white in the image.

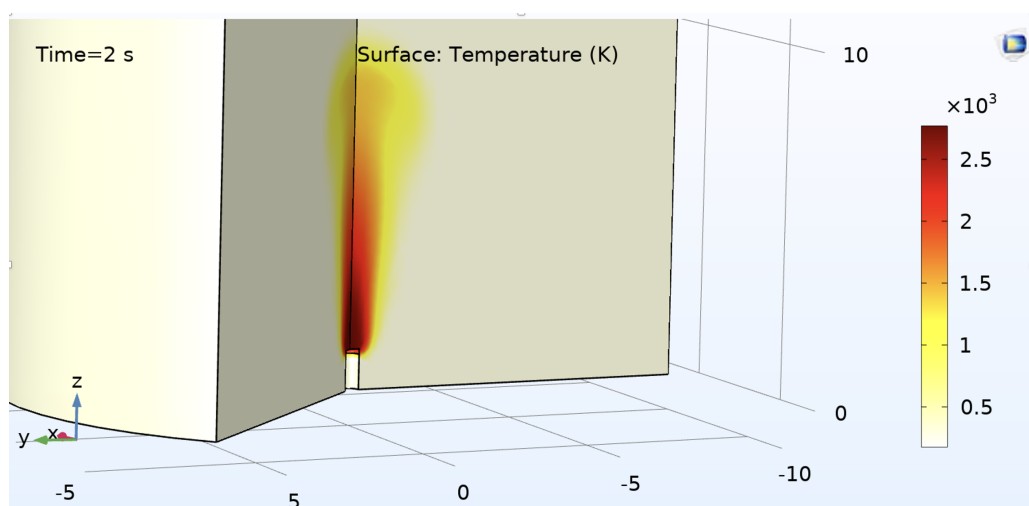

**Figure 1.** Image showing the COMSOL 3D modeling completed for this study. The thermal radiation away from the simulated barrel (white area) was estimated for multiple cases.

### 2.2.2. Pressure and Debris Modeling Details

The Integrated Violence Model or IVM is a thermochemical-based model that tracks the temperature, pressure, composition, and enthalpy of gases inside a structure as a function of time as gases enter (from burning propellant) and exit the structure (through dynamic vents). Material and energy balances of the gases inside the structure are solved each explicit time-step with the following assumptions: reversible process, ideal gas, immediate thermal equilibrium in the gas volume, and uniform pressure throughout the control volume. Material and energy balances are, specifically:

$$\frac{dn}{dt} = \dot{n}_{in} - \dot{n}_{out} \tag{3}$$

$$\frac{dnU}{dt} = \dot{n}H_{in} - \dot{n}H + q_{gen} \tag{4}$$

where $t$ is time, $n$ is moles, $U$ is internal energy, and $H$ is enthalpy. The dot represents a rate, and the $q_{gen}$ term accounts for the afterburning of reactive gases such as carbon monoxide or hydrogen. Afterburning was assumed to be 1:1 in that for every mole of gases reacting with oxygen, 1 mole of product gases is produced so that there is no change in the number of moles in the chamber. This is conservative, as usually 2 moles of CO or H2 (common combustion products that participate in afterburning) react with 1 mole of oxygen to form 2 moles of products. Thus, 3 reactant moles typically result in 2 product moles. The $q_{gen}$ term is thus:

$$q_{gen} = k_a \cdot P_{CO} \cdot P_{O2} \cdot H_{comb} \tag{5}$$

where $k_a$ is the burn constant (units of mol per second per pressure squared), $P_{CO}$ is the partial pressure of the reactant gases (such as CO), $P_{O2}$ is the partial pressure of the oxidizing gases (such as oxygen), and $H_{comb}$ is the combustion heat of reacting gases (CO) afterburning with the O2 in the air inside the chamber. Four gases are modeled to represent the many different gases: inert gases (modeled as carbon dioxide), reactive gases (modeled as carbon monoxide), oxidizing gases (modeled as oxygen), and water vapor.

The amount of gas generated by the propellant depends on the burn rate and the gas pressure according to:

$$\dot{n}_{in} = -g_{gen} \cdot \frac{dm_s}{dt} = -k \cdot \left( \frac{P}{P_{ref}} \right)^{\beta} \tag{6}$$

where $g_{gen}$ is the gas generation rate, $m_s$ is the mass of the 1.3 substance, $k$ is the atmospheric burn rate, $P$ is the gas pressure, and $\beta$ is the pressure exponent. Other gas generation models

can be used to simulate dust, vapor explosions, or burn rates that depend on the mass of substance present.

Gases exit the space through vents of various sizes that can open as a door or as a panel with the exit mass flow rate depending on the Mach number, as outlined in gaseous flow equations such as those outlined in Perry's Chemical Engineering Handbook Seventh Edition, McGraw-Hill 1997, page 6–22 and 6–23.

Once the derivative of the temperature and moles are known (from the above equations), the gaseous temperature and number of moles in the structure can be found by solving those differential equations (solved here with an explicit fourth-order Runge–Kutta approach). The pressure can also be found using the ideal gas relationship. Knowing the pressure as a function of time inside the structure can indicate whether the venting is sufficient to prevent structural failure.

Inputs to the model include the atmospheric burn rate of the propellant, burn rate pressure exponent, amount of gas generated per mass, temperature or enthalpy of the gas generated, composition of the combustion gases, total gas volume of the structure, and vent sizes and the associated vent dynamics (weight and opening type such as a door or panel). With these relatively few parameters, accurate pressure versus time results have been obtained for a number of scenarios with burning propellant including small pressure vessels, processing dryers, and concrete structures. Example simulations of the internal pressure and temperature in several different conditions with the Integrated Violence Model are published online at https://ivm13.com/, accessed 23 August 2023.

In addition to determining whether there is sufficient venting to prevent extreme pressures inside the structure, IVM was used to predict the debris throw distances when the pressure does exceed the strength of the structure. The size distribution or number of fragments for each fragment size (normalized by the total mass) was used for concrete as obtained from the Kasun structure testing in Reference [3]. Those debris fragments were then randomly positioned such that the concrete structure was roughly reproduced. The initial direction for each fragment was the vector from the geometric center of the structure to the initial fragment location. Then, those fragments were accelerated with the venting pressurized gases, with that acceleration decreasing as the fragments moved away from their initial positions. The debris trajectories for each fragment are based on that acceleration and initial trajectory from their position relative to the center of the structure. Drag forces are included, as is gravity. The max fragment energy total from all the debris is held to 5 percent of the product of the failure pressure and void volume of the structure. Once the debris fragment strikes the ground, it is assumed to stay there. With this approach, the fragment distribution after the event was found with surprisingly good agreement with the experimental fragment distributions, as reported in [5]. This same approach was used to predict the hazardous fragment ranges for larger, untested concrete structures.

## 3. Results

Results of the literature and modeling analysis of confined and unconfined scenarios are summarized here. The existing HD 1.3 IBD distances are compared to the calculated separation distances to prevent second degree burns and prevent exposure to hazardous fragment densities greater than 1 in 600 square feet. For both confined and unconfined scenarios, a new curve is recommended for safe and efficient work with HD 1.3 ammunition and explosives.

### 3.1. Comparison of ESSD to IBD for Unconfined Scenarios

Table 1 lists literature values of more than 100 open burn tests with HD 1.3 substances and a total of over 600 heat flux measurements. From these data, ESSDs have been estimated per the above method. The distances for each mass were then compared with the current HD 1.3 IBD separation distances. The resulting data-points of the safe separation distances to prevent second degree burns are given in Figure 2.

As can be observed in Figure 2, the current separation distances (blue line) for HD 1.3 substances in the DESR 6055.09 protects against second degree burns when those substances are unconfined, as almost all of the points are under that blue line. In the cases where ESSDs are above the IBD distance, the test data are likely overly conservative, as the burn times from those events were not reported in the literature and thus were estimated (see References [11,22] and the csv file at the github site). Moreover, for some of the literature values, the peak heat flux is reported, but the average heat flux for the duration of the burn was not reported. In those cases, the average was assumed to be half the peak value.

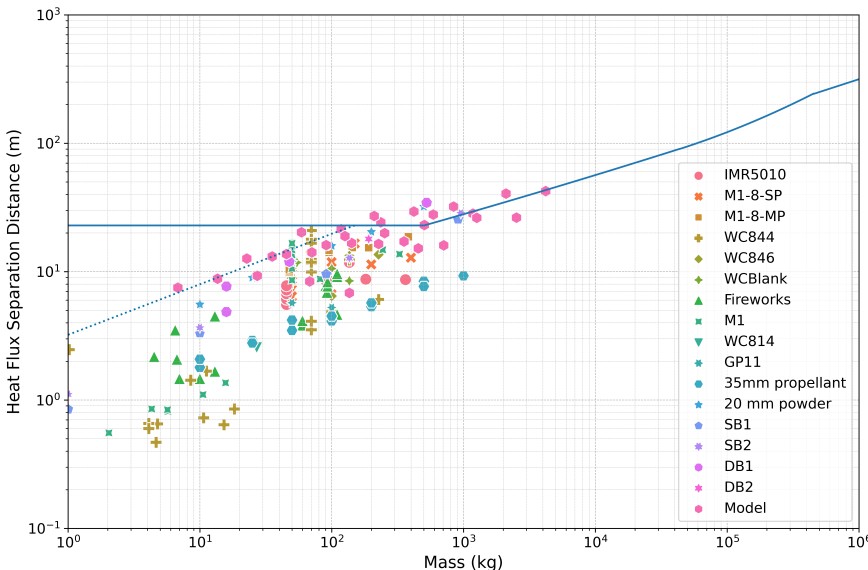

**Figure 2.** Plot of the unconfined explosives' safety separation distances from testing and modeling, as reported in References [6–9,13]. The solid blue line is the current HD 1.3 IBD per DESR 6055.09. The dotted blue line is the proposed adjustment for smaller masses (<145 kg). The data-point colors and symbols are specific to the given propellant type, most of which are different types of smokeless powder, as described in Table 1 and the respective references. Almost all of the estimated ESSD distances from the test data are within the current IBD, indicating sufficient protection against potential injury for the unconfined burning of energetic material.

The model points shown in Figure 2 are from the COMSOL modeling (Ref. [13]) of radiative heat transfer from a source with different diameters and propellant burn rates according to the modeling method described above. Some of the data from the model are slightly above the IBD line for scenarios with high burn rates. Those points are slightly above the line, as the gas temperature is likely too high in the model (2800 K); more realistic temperatures are in the range of 1000–2100 K (see, for example, the fireball temperatures from MTV flares of 1700–1800 K, as reported in Reference [14]).

The above results and comparison were for unconfined scenarios (open burning). Confined events of HD 1.3 substances and articles can have much more violent results, as presented in the next section.

### 3.2. ESSD for HD 1.3 Confined Scenarios

Confined scenarios where rapidly produced hot gases are not allowed to sufficiently escape can result in a pressurization and violent bursting of the structure. Typically, that violence can scale with the product of the failure pressure and the internal gaseous volume, as is true for pressurized cylinders. Those pressure bursts can generate fragments or debris that can travel long distances in addition to generating a large fireball.

Literature values for multiple confined scenarios are summarized graphically in Figure 3. The confinement where the venting is limited is in concrete structures (References [3,21]), earth-covered structures (Reference [18]), and a very large rocket motor (References [19,20]). Some of the test data for the confined scenarios did not have measured

heat flux values, but the fireball size was reported or estimated from a video. The explosives' safety separation distance was then found per the above enumerated steps, assuming an average heat flux of 80 kW/m² at the fireball distance. The value of 80 kW/m² is reported by Dorofeev at the surface of a fireball (Ref. [23]). That value is considered conservative, as the predicted distance for the allowed heat flux is so far away from the fireball edge. The accuracy of the point source model is relied upon with a power law dependence on distance of two. In reality, it is likely to be greater than two, due to the atmospheric absorbance of the radiated energy. For example, 2.09 is used in some NATO combustion models [24].

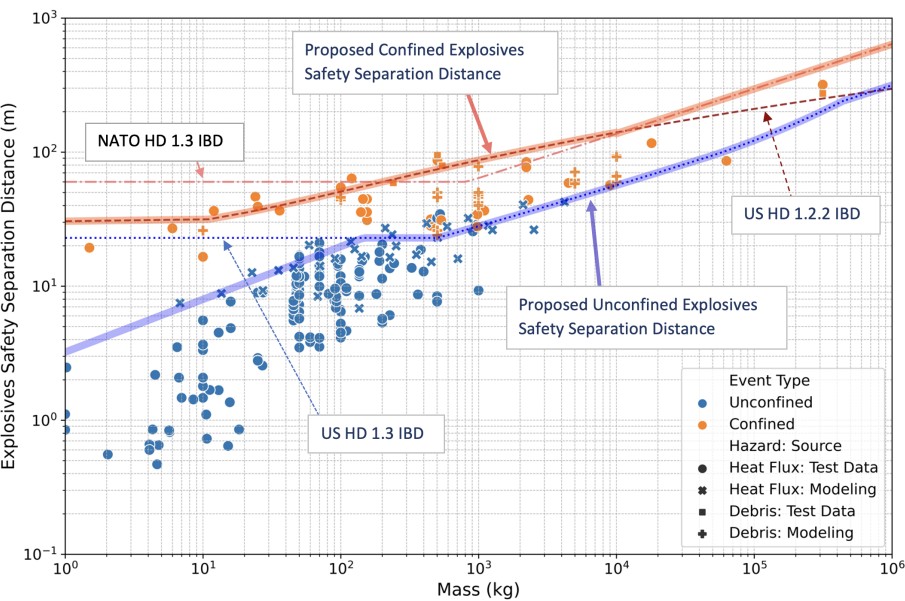

**Figure 3.** Plot of the data (as points) for the unconfined (blue) and confined (orange) scenarios. The unconfined data points are repeated from Figure 2. Also shown are the proposed ESSDs (as lines). The recommended unconfined ESSDs (thick blue line) and confined ESSDs (thick orange line) are as per Table 2.

The debris estimates for the confined scenarios are primarily from the Kasun structure testing reported in References [3,21]. A significant amount of work was completed to catalog and mark all of the debris from multiple tests. The Hazardous Fragment Distance (HFD) of 1 hazardous fragment in 600 square feet was then determined. Integrated Violence Modeling was completed for multiple scenarios, as outlined above, where the fragment distances and densities were predicted from the test parameters given the concrete structure fragment size distribution, as described above (Ref. [5]). For the Titan rocket motor event [20], the HFD was estimated as 0.6 times the maximum fragment distance (457 m for a mass of more than 300,000 kg) estimated from video evidence.

The relations for the recommended separation distances that are intended to prevent second degree burns and exposure to hazardous fragment densities greater than 1 in 600 square feet are given in Figure 3 for both unconfined and confined scenarios. The equations used to generate those lines are given in Table 2 in SI units and in Table 3 for English units.

The relationships given in Tables 2 and 3 were not found from a fit to the max of the data but are based on relationships previously defined (such as the US HD 1.3, US HD 1.2.2 IBD, and NATO HD 1.3 IBD) that appear to match the upper values of the data and modeling results.

**Table 2.** Proposed HD 1.3 Explosives Safety Separation Distances (SI Units).

| Condition | Mass, M (kg) | Relation (m) |
|---|---|---|
| Unconfined | $M \leq 453$ | $\min(3.216 \cdot M^{0.3939}, 22.9)$ |
|  | $453 < M \leq 43{,}540$ | $\max(22.9, \exp[1.4715 + 0.2429 \cdot \ln(M) + 0.00384 \cdot (\ln(M))^2])$ |
|  | $43{,}540 < M \leq 453{,}590$ | $\exp[5.5938 - 0.5344 \cdot \ln(M) + 0.04046 \cdot (\ln(M))^2]$ |
|  | $453{,}590 < M$ | $3.17 \cdot M^{1/3}$ |
| Confined | $M \leq 10{,}000$ | $\max(30.5, 28.127 - 2.364 \cdot \ln(M) + 1.577 \cdot \ln(M)^2)$ |
|  | $10{,}000 < M$ | $6.4 \cdot M^{1/3}$ |

**Table 3.** Proposed HD 1.3 Explosives Safety Separation Distances (English Units).

| Condition | Mass, M (lb) | Relation (ft) |
|---|---|---|
| Unconfined | $M \leq 1000$ | $\min(7.73 \cdot M^{0.3939}, 75)$ |
|  | $1000 < M \leq 96{,}000$ | $\max(75, \exp[2.463 + 0.238 \cdot \ln(M) + 0.00378 \cdot (\ln(M))^2])$ |
|  | $96{,}000 < M \leq 1{,}000{,}000$ | $\exp[6.8633 - 0.5418 \cdot \ln(M) + 0.03828 \cdot (\ln(M))^2]$ |
|  | $1{,}000{,}000 < M$ | $7.99 \cdot M^{1/3}$ |
| Confined | $M \leq 22{,}050$ | $\max(100, 101.61 - 15.926 \cdot \ln(M) + 5.173 \cdot \ln(M)^2)$ |
|  | $22{,}050 < M$ | $16.13 \cdot M^{1/3}$ |

## 4. Discussion

We have recommended updated ESSD values for HD1.3 substances and articles based on a large amount of test data that would allow for more material at current separation distances when the unconfined quantity is less than 145 kg. Additionally, the recommended values at other masses would better protect people and property for confined scenarios and are more consistent with the NATO distances used around the world. The difference between the confined and unconfined cases are currently subjective in that a required vent area limiting the internal gas pressure in the building or structure is not defined for all possible structures. Further work to readily define the difference between the two is needed.

The recommended ESSDs used the average heat flux for the burn duration. Other methods could be used to estimate the separation distance such as using the peak heat flux and a burn duration that also conserves energy. Doing so may yield slightly higher explosives safety separation distances. The average heat flux was chosen here due to the layers of conservatism including the inverse squared law relating the radiation to distance (more likely a faster decline) and the exposure time to prevent second degree burns in Equation (1) which is also conservative.

The recommended ESSDs are not directional. In other words, the ESSDs are reported in the direction where the largest heat flux would be experienced. Directional effects could be accounted for in a straightforward way, for example, by including the location information of vents through which the hot gases exit and afterburn. Similarly, instead of using the net explosive weight (NEW) as the key parameter, the ESSD could be related to a different set of key parameters such as the burn rate, loading density, vent area, and vent area location, among others. The net explosive weight was used here, as the current US and NATO ESSDs are functions of the NEW.

## 5. Conclusions

A combined analysis of hundreds of HD 1.3 tests, accidents, and detailed modeling results show that there is a significant difference in the heat flux and debris hazards between confined and unconfined scenarios. Additionally, the data analysis shows that current explosives' safety separation distances for HD 1.3 ammunition and explosives do adequately protect people from hazards for unconfined scenarios. However, hazards from confined scenarios are currently not adequately protected against. Those hazards to personnel include when the internal pressure rapidly rises from insufficient venting

and the structure fails violently or when the gases from the structure are violently forced out, resulting in much larger than expected directional heat fluxes. The recommended unconfined and confined explosives' safety separation distances given here would protect people adequately for both scenarios.

Additionally, the recommended adjustment to the existing HD 1.3 curve for unconfined scenarios has efficiencies due to lower separation distances required with masses less than 145 kg, as the current HD 1.3 IBD relation appears to be overly conservative at low masses. Both modeling and experimental data have proved valuable in evaluating and recommending the explosive safety separation distances for unconfined and confined scenarios with HD 1.3 ammunition and explosives.

**Author Contributions:** Conceptualization, C.G., M.L. and J.C.; methodology, C.G.; formal analysis, C.G.; data curation, C.G.; writing—original draft preparation, C.G.; writing—review and editing, C.G. and J.C. All authors have read and agreed to the published version of the manuscript.

**Funding:** This research received no external funding.

**Data Availability Statement:** The compilation of data together with the reference and calculations are available in a public github repository (github.com/clint-bg/publicationdata/safeSeparation).

**Acknowledgments:** We acknowledge helpful discussions with Martijn van der Voort and Matt Ferran with the Munitions Safety Information Analysis Center (MSIAC) and for providing some of the referenced heat flux data points.

**Conflicts of Interest:** The authors declare no conflict of interest.

## Abbreviations

The following abbreviations are used in this manuscript:

| | |
|---|---|
| $m, M$ | Mass of energetic substance, kilograms |
| $d$ | Distance from substance, article, or structure, meters |
| $t$ | Time, seconds |
| $q$ | Heat flux, kilowatts per square meter |
| $\alpha$ | Point source model parameter, kilowatts |
| $n$ | Moles inside the structure |
| $g_{gen}$ | Gas generation rate, mol/kg |
| $P$ | Pressure, Pa |
| $k$ | Rate constant or atmospheric burn rate |
| $H$ | Enthalpy, J/mol |
| $U$ | Internal energy, J/mol |
| $q_{gen}$ | Heat generation from afterburning, W |

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
