# Peer review of "Recommended Separation Distances for 1.3 Ammunition and Explosives"

_fire, doi:10.3390/fire6090331_

Round 1

Reviewer 1 Report

This manuscript addresses the issue of the current separation distances for Hazard Division (HD) 1.3 substances and articles used in the United States, in some cases, may not adequately protect against the effects from heat flux and debris when those substances and articles are ignited in a confined structure. The author calculated the results through a large number of experiments and compared them with existing reports, ultimately obtaining a more recommended safety distance. The research content of the manuscript is more focused on solving practical engineering problems. The manuscript does need major revision to address my comments below.

1: Add some key contents of HD1.3 (1.1, 1.2 if needed) in the introduction.

2: Provide additional explanations for the codes in the legend (Figure 2) at the bottom or in the figure for a more intuitive observation of the information in the figure. (M*/GP*/DB*/)

3: Besides net explosive weight, does the equivalent of explosives need to be considered? Or other major factors?

4: The manuscript lacks in-depth analysis and discussion of the experimental results, especially at the mechanism level. It is recommended to supplement them appropriately.

5: Whether the EEDS data reported in various literature have been normalized. Suggest explaining in the manuscript.

Reviewer 2 Report

This paper looks at the ESSD from the potential hazardous effects from propellants, explosives, and pyrotechnics and analyzed the heat flux and debris hazards between confined and unconfined scenarios according to experimental and theoretical models.The content are novel and interesting enough. There is a question that how to determine the unconfined quantities less than 145 kilograms.
